# Data Augmentation Effects on Highly Imbalanced EEG Datasets for Automatic Detection of Photoparoxysmal Responses

**DOI:** 10.3390/s23042312

**Published:** 2023-02-19

**Authors:** Fernando Moncada Martins, Víctor Manuel González Suárez, José Ramón Villar Flecha, Beatriz García López

**Affiliations:** 1Electrical Engineering Department, University of Oviedo, 33203 Gijón, Spain; 2Computer Science Department, University of Oviedo, 33003 Oviedo, Spain; 3Neurophysiology Department, University Hospital of Burgos, 09006 Burgos, Spain

**Keywords:** electroencephalography, EEG, Photoparoxysmal Response, PPR, Machine Learning, Data Augmentation, photosensitivity, epilepsy

## Abstract

Photosensitivity is a neurological disorder in which a person’s brain produces epileptic discharges, known as Photoparoxysmal Responses (PPRs), when it receives certain visual stimuli. The current standardized diagnosis process used in hospitals consists of submitting the subject to the Intermittent Photic Stimulation process and attempting to trigger these phenomena. The brain activity is measured by an Electroencephalogram (EEG), and the clinical specialists manually look for the PPRs that were provoked during the session. Due to the nature of this disorder, long EEG recordings may contain very few PPR segments, meaning that a highly imbalanced dataset is available. To tackle this problem, this research focused on applying Data Augmentation (DA) to create synthetic PPR segments from the real ones, improving the balance of the dataset and, thus, the global performance of the Machine Learning techniques applied for automatic PPR detection. K-Nearest Neighbors and a One-Hidden-Dense-Layer Neural Network were employed to evaluate the performance of this DA stage. The results showed that DA is able to improve the models, making them more robust and more able to generalize. A comparison with the results obtained from a previous experiment also showed a performance improvement of around 20% for the Accuracy and Specificity measurements without Sensitivity suffering any losses. This project is currently being carried out with subjects at Burgos University Hospital, Spain.

## 1. Introduction

Photosensitivity is a neurological condition in which the patient’s brain responds abnormally to certain visual stimuli, such as light reflections, flashing lights, or intermittent patterns. These anomalous responses are called Photoparoxysmal Responses (PPRs) and take the form of different types of electrical epileptic discharges, which can differ in intensity or spread throughout the brain. In this respect, the study published in [1] proposed a taxonomy of four different PPRs types:Type-1: spikes in the occipital region.Type-2: spikes followed by a biphasic slow wave in the occipital and parietal regions.Type-3: spikes followed by a biphasic slow wave in the occipital and parietal regions, spreading to the frontal regions.Type-4: generalized poly-spikes and waves.

The higher the type’s number, the more dangerous the photosensitivity is, with the Type-4 PPR corresponding to the most-severe case. Generally, PPRs are epilepsy manifestations that can result in generalized seizures; however, only about 6% of epileptic people suffer from photosensitivity and PPRs [2] and their causal relationship with the abrupt changes in visual stimulation characteristic of video games [3], advertisements [4], etc. The exposure of photosensitive and epilepsy patients to this kind of digital content and environments has become a major concern with the proliferation of video games and virtual reality technologies in recent years [5].

Diagnosing a patient’s photosensitivity follows a standardized clinical procedure known as Intermittent Photic Stimulation (IPS) [6,7,8,9]. This procedure alternates resting and stimulating periods of time; during a stimulation period, a white flashing light switches on and off at a given frequency. The flashing frequency gradually increases from a minimum to a maximum and, then, gradually decreases back to the minimum again. Whenever a PPR is detected, the current increasing or decreasing process is halted to avoid any onsets, diagnosing the patient with photosensitivity and determining the photosensitivity frequency limits. The neurophysiologists detect the PPRs using the Electroencephalogram (EEG) signals from the patient, monitoring these signals during the whole procedure. Figure 1 shows an example of two PPRs obtained while flashing a patient.

This procedure has two main drawbacks: On the one hand, whether the white flashing stimulation is enough to detect all the types of PPR is arguable [10]. On the other hand, the amount of time required for a complete analysis of the EEG recordings is too large, reducing the efficiency of the whole procedure and the performance of the neurophysiologist unit. Furthermore, the EEG recording in Figure 1 shows the impossibility of labeling a PPR as belonging to one single type because all the PPR types usually merge within a single PPR event. Moreover, external variables, such as medical treatment, sleep quality, or even the time of the day, may introduce variations in the EEG signals for a PPR event.

As a result, the EEG recordings from IPS sessions show several issues. Firstly, the morphology and features of the brain activity (including PPRs) may significantly change between different patients and even between different EEG sessions of the same patient due to his/her ongoing condition [11]. It is well known that the resting brain activity of the patients also presents remarkable differences according to their ongoing conditions, introducing noise in the process and difficulties in PPR detection. Finally, obtaining representative data from PPRs becomes a challenging task not only because of the small percentage of the affected population [2,12,13], but also because the number of PPRs within a single IPS EEG recording is fairly small as long as the procedure stops once a PPR is found. As a consequence, gathering a suitable dataset for training and testing Artificial Intelligence (AI) and Machine Learning (ML) models is always in compromise: in the normal scenario, these models deal with an extremely unbalanced dataset.

To our knowledge, the literature concerning automatic PPR detection only includes a handful of studies, showing that this problem has not received the focus of the research community. For instance, there are authors that claim that using the IPS procedure is not suitable for PPR detection, proposing using flashing at high frequencies instead [10]. The study in [14] discriminated between normal or PPR stimulation regions by means of the aggregation of the Fourier components calculated on sliding windows, whose lengths varied according to the flashing frequency. Besides, the studies in [15,16] were the only ones that proposed ML for automatic PPR detection. There could be room for transfer learning from research in closely related fields, such as those focused on generalized epilepsy seizure detection. We can cite some examples, such as the research in [17], which proposed using Extreme Gradient Boosting for seizure classification, or the study in [18], which made use of the fluctuation of the EEG channel higher and lower frequencies; the use of the Permutation Rényi Entropy for differentiating interictal and ictal states was proposed in [19]. Simple ML techniques such as the Artificial Neural Network or K-Nearest Neighbors algorithm were also applied in the automatic detection of ictal discharges and inter-ictal states [20], while a more complex Deep Learning (DL) method such as a channel-independent Long Short-Term Memory Network was proposed in [21]. Other studies made use of different clinical equipment, bio-markers, and biomedical measures for the same purpose, such as Electrocardiography (ECG) [22,23,24], ECG combined with EEG data [25], electromyography [26,27], or magnetoencephalography [28]. The existence of such recent studies on a subject that has been so widely studied over several decades clearly shows that the detection and prediction of epileptic seizures are far from being solved [29].

Finally, the dataset imbalance problem can be solved by creating synthetic data from the real dataset using Data Augmentation (DA) algorithms. Since recording and finding the most-suitable clinical EEG recordings from real patients can take much time and effort by neurophysiologists, DA represents a very useful technique that grows the dataset with realistically and artificially created EEG PPR instances resembling the real ones; therefore, DA aims to balance the dataset and, thus, the capacity of obtaining competitive models.

DA has always been more applied to image processing applications, but currently, it is gaining notoriety in Time-Series applications, where it includes operations from very simple ones—such as jittering, warping, or slicing—to more advanced techniques such as the application of Generative Networks [30]. It has been widely applied in classification tasks with DL models, such as forecasting problems [31,32] or medical issues, such as the monitoring of medication tampering in [33] using multivariate time signals, the introduction of a DA stage in the training of an LSTM-based DL model applied to fall detection in [34], or the improving of ECG classification in [35].

Due to the complex patterns present in EEG signals, DL algorithms often perform better in their analysis because they can learn such patterns in greater depth, but a large-scale balanced dataset is needed, so DA is a fairly common solution to improve a clinical dataset: [36,37] collected many recent studies that applied some kind of DA process to improve their DL performance. Even for the same task, different DA approaches can be used to create the synthetic data as in the case of emotion recognition, where [38] decided to apply Convolutional Neural Networks (CNNs), while [39] used Deep Generative models. The work of [40] applied and compared the effect of six different DA techniques on the classification using a CNN from the most-simple ones (averaging or segment recombination) to more refined ones such as autoencoders. A DL neural network that includes a combination of Data Augmentation and Domain Adaptation modules was proposed in [41] to improve the accuracy of single-channel EEG classification.

In this study, we focused on reducing the effects of the imbalanced nature of the dataset. To do so, we suggest the use of a very specific Cross-Validation stage complemented by an additional DA step. While the designed Cross-Validation scheme tries to reduce the percentage of training samples from the majority class, the DA step aims to balance the training dataset by generating artificial PPR instances by mixing EEG windows belonging to the minority class. To evaluate the performance of these improvements, we propose using the two-best PPR detection algorithms from our previous experiments: a two-class K Nearest Neighbor classifier and a Dense-Layer Neural Network.

The structure of this paper is as follows: The next subsection gives some context to this research. Section 2 focuses on the detailed description of the materials and methodologies applied in the study. Section 3 presents the results of the experimentation and their analysis. The last section contains the conclusions of this research.

### Virtual Reality and Artificial Intelligence for PPR Detection

This study is part of a larger project that proposes the introduction of Virtual Reality (VR) and AI technology for the photosensitivity diagnosis and evaluation process, as a supporting tool for neurophysiological specialists [15]. We call the integration of these two fields virtual-reality-enhanced Artificial Intelligence (**vAI**), with VR offering the possibility of recreating real-world scenarios in a virtual environment, while AI algorithms are able to perform real-time analysis of EEG recordings, automatically detecting the abnormal brain activity, such as PPRs, reducing the time required by the neurophysiologists for this task. In Figure 2, a scheme of how the IPS system works currently versus how it would work with vAI is shown. New protocols can be designed using VR, including the development of training tools for the patients to learn how to distinguish dangerous scenarios. AI is responsible for assessing the level of stimulus to keep the patient safe.

## 2. Materials and Methods

Figure 3 shows the sequence of stages in this study; this section describes each of these stages. Firstly, Section 2.1 describes the dataset used in this study, giving the details on the data recording process. Secondly, Section 2.2 focuses on the designed DA approach for dataset balancing. Afterward, Section 2.3 introduces the preprocessing of the EEG signals, the feature extraction, and the dimensional reduction stages. The Cross-Validation stage introduced in this research is thoroughly defined and explained in Section 2.4. Finally, Section 2.4 focuses on the ML modeling techniques for classifying each EEG window as a PPR or a normal window.

### 2.1. Dataset

The clinical neurophysiologists from the Neurophysiology Service at Burgos University Hospital recorded and annotated the dataset used in this research. This dataset included data from 10 patients diagnosed with different degrees of photosensitivity. Each patient went through an IPS session, recording the EEG channels. The equipment in the facilities included a Natus Nicolet v44 for EEG recording and a Natus Neuroworks9 for real-time EEG visualization. The data were anonymized, so no extra information—such as gender or sex—was kept, following the hospital’s privacy protocol for the essay.

Each session consisted of a 3- to 5-min continuous recording using an EEG cap while the patient was stimulated by applying the first half of the IPS procedure, corresponding to the ascending standard frequencies from 1 Hz up to 50 Hz. The sampling rate for the EEG signals was 500 Hz; placing up to 19 electrodes according to the 10–20 standardized system [42] (see Figure 4). The EEG recordings were analyzed considering the EEG average montage, which is the one that clinical neurophysiologists use on a daily basis. This average montage computes the global average of the values among all the EEG channels; then, each sample is modified by subtracting this calculated global average value. One-second intervals are marked in this montage, representing the standard time slot used in the EEG analysis.

To annotate the EEG recordings, the clinical specialists visually analyzed all raw EEG recordings, marking the starting and ending points of any triggered PPR during the stimulation. PPR phenomena can present a very variable duration (e.g., one may be triggered for only one-tenth of a second, while the next can last up to five seconds straight). As mentioned before, the montage has one-second intervals to serve as a guide for the clinical specialists, so the EEG recordings were split using a 1-second length sliding window with 90% overlapping. Then, each window was manually labeled as PPR or non-PPR according to whether or not it included part of a PPR interval. Before applying Data Augmentation, the balance of the original dataset was as follows:Total number of EEG windows: 29,190:–Number of non-PPR windows: 27,968 (**95.81%**).–Number of PPR windows: 1222 (**4.19%**).

As can be observed, the dataset showed an extreme imbalance, negatively affecting the learning capacity of the models. This research proposes a DA strategy to tackle this problem, as explained in the next section.

### 2.2. The Data Augmentation Strategy

The DA stage aimed to generate realistic new PPR windows that clearly resemble actual PPR ones. Using the DA stage increases the number of PPR windows, introducing more representativeness to the training dataset. For the purposes of this research, we define an ad hoc method for DA that merges two actual PPR windows. The idea is to split the selected windows into **n** intervals, generating a new PPR window by collecting alternating intervals from each parent, a similar approach to the recombination method applied in [40].

Nevertheless, we must consider when and where a PPR appears within a PPR window; four different possibilities arise: (i) the PPR window contains the starting part of a PPR event; (ii) the PPR window contains the final part of a PPR event; (iii) the PPR window represents part of a PPR event; (iv) the PPR window contains the whole PPR event (see Figure 5, which depicts these four cases). The DA stage must merge PPR windows from the same case to produce better and more realistic synthetic windows. Otherwise, the obtained EEG segments may become meaningless from a neurophysiological point of view. Additionally, the DA also balances the dataset in terms of the number of PPR windows from each case, so the total number of new instances is the same for each of them.

Therefore, in creating a new realistic synthetic PPR window *C*, the DA first selects the group to extend (one of the four possibilities mentioned before), choosing two random actual PPR windows *A* and *B* from the dataset. Besides, the number of cut-off points *n* is defined in advance, dividing both *A* and *B* windows into n+1 segments of the same length (in this study, we kept n=3, so windows *A* and *B* were cut by three cut-off points located every 125 samples (at Samples 125, 250, and 375), thus generating 4 segments of 125 samples in length each).

The new synthetic PPR window *C* is then compounded with alternating segments, one from each parent *A* and *B*, as shown in Figure 6: starting with the first segment from *A*, then the second segment from *B* is added, then the third segment from *A* again, and so on. Equation (Equation 1) shows the general sequence to produce the synthetic PPR window from its parents for the current *n* parameter.
(1)$$C=\left\{A_{0},B_{1},A_{2},B_{3},...,A_{i-1},B_{i},A_{i+1},...\right\}\to i\in \left[0,n\right]$$

Besides, abrupt EEG signal changes need to be avoided when merging two PPR windows because they do not occur; these abrupt changes may be due to the high difference between the ending value from one segment and the starting value in the next segment arranged in a sequence. In this study, the use of interpolation around the cut-point helps in keeping the signal’s continuity, providing smooth transitions among segments and avoiding these sudden value changes. Equation (Equation 2) shows the proposal for the interpolation, where *x* is the cut-point and when the left segment is from *A* and the right segment from *B*; obviously, the formula would be implemented in a reverse way. Furthermore, Figure 6 illustrates the generation and smoothing of the changes; when more than one EEG channel is considered, the same cut-points are used among all the channels to keep coherency.
(2)$$\begin{array}{cc} C(x-2) & =\;A(x-2)\\ C(x-1) & =0.75*A(x-1)+0.25*B(x-1)\\ C(x) & =\;0.5*A(x)\;+\;0.50*B(x)\\ C(x+1) & =0.25*A(x+1)+0.75*B(x+1)\\ C(x+2) & =\;B(x+2) \end{array}$$

Interestingly, the new synthetic EEG windows have a very similar spectrogram to that of the parents. Figure 7 shows the spectrograms of a DA generation, with two random parents from one of the groups and the synthetic sibling. Some spectrograms’ changes appear in the range from 50 to 100 Hz, although the main distribution of the signals is similar in both the parents and the siblings. We assumed that the observed changes are not meaningful, which means that the cut points and the interpolation proposed in this research do not introduce any disturbances to the TS. Nevertheless, the subsequent pre-processing stages filter these differences.

### 2.3. Preprocessing and Dimensional Reduction

After applying DA to the dataset, preprocessing, feature extraction, and dimensional reduction take place (see Figure 8). Despite recording 19 EEG channels, neurophysiologists only consider five of them to detect PPRs: Fz, F2, F4, O1, and O2. In a previous study [15], we analyzed this subset of channels and concluded that **Fz** was the most-plausible channel for PPR detection if only one channel were to be selected; therefore, for this study, we used only this Fz channel. Future work will consider using more EEG channels for PPR detection to improve the performance of the models. The preprocessing of an EEG window includes removing the average and applying a Notch filter at 50 Hz plus a band-pass filter in the range of 1 to 50 Hz.

Up to 31 features from different domains were calculated using the library TSFEL [43]; Table 1 lists the domains and transformations considered, with the corresponding study where the transformation was originally defined. These features were scaled to the interval [0.0,1.0], and Principal Component Analysis (PCA) [44] was performed; the PCA components representing up to the 95% of the variability in the data were preserved, leading to a new domain of 12 features. It is worth mentioning that we also performed dimensional reduction using Independent Component Analysis (ICA) and Locally Linear Embeddings (LLEs); however, the results obtained through PCA represent the best solution bed.

### 2.4. Cross-Validation Scheme

Applying DA is not enough to counterbalance the high imbalance character of the dataset: the percentage of synthetic PPR windows would be too high, biasing the learning process. Therefore, the number of non-PPR instances must also be reduced, so that the combined action of the two methods produces a balanced dataset. This reduction in the number of non-PPR windows should be carefully performed so that no bias is introduced in this subsampling process as well.

In a preliminary study, unsupervised learning was used to group the non-PPR windows, aimed to find some structure in the data. Unfortunately, independent of the clustering technique, the great majority of the non-PPR windows were grouped in the same big cluster. Instead, sub-sampling with replacement is proposed for the reduction in the number of non-PPR windows, as depicted in Figure 9. Thus, each non-PPR window’s fold includes randomly selected non-PPR windows. Interestingly, a non-PPR window can be selected for more than one single fold: in this way, we tried to keep the variability and heterogeneity in all the different folds generated in the Cross-Validation process. Probabilistic sub-sampling with replacement could have been used as well, reducing the probability of a non-PPR window once chosen in a fold; however, for the sake of simplicity, we kept the normal sub-sampling with replacement.

Each final training fold includes a balanced number of PPR, either actual or synthetic, and non-PPR windows. Furthermore, two test datasets were generated: with and without DA-generated windows. These two test sets were created for comparison purposes of the models’ performances using only real data and a combination of real data and the DA instances. This performance comparison would help in understanding how the DA stage affects each of the models. Figure 9 shows the procedure for the generation of the training and test sets, following the next steps:There were 3000 non-PPR and 500 PPR windows randomly selected.There were 2500 synthetic PPR windows created from the 500 selected, up to a total of 3000 PPR instances.The training set was formed by these 6000 instances with a perfect balance of 50%.Two test sets were formed:The first one (Test 1) was made from the rest of the non-PPR and only the real PPR windows.The test synthetic PPR windows were created up until another 3000 PPR instances again.The second test set (Test 2) was created from the rest of the non-PPR and the synthetic PPR windows.

### 2.5. Modeling and Evaluation

Once the training and test sets were created, the training and the evaluation of the ML classifiers were performed. The algorithms selected for this study were the best techniques found in our preliminary research [16]: two-class K-Nearest Neighbors (2C-KNN) and a Neural Network with a Dense Layer as the hidden layer (DL-NN). We considered other ML models, such as SVM or Random Forests [15]; however, the performance of the 2C-KNN and the DL-NN were better than with other methods. For each model, different parameter values were also tested: for 2C-KNN, the *K* number of neighbors tested was {3, 5, 7, 9, 11, 13, 15}; for DL-NN, the *N* number of hidden neurons tested was {10, 20, 30, 40, 50}.

To measure the performance of the ML techniques for the PPR detection, the Accuracy (ACC), Sensitivity (SENS), and Specificity (SPEC) measurements were calculated according to Equations (Equation 3)–(Equation 5), respectively, where *TP*, *TN*, *FP*, and *FN* are the True Positive, True Negative, False Positive, and False Negative classification values, respectively; $N_{data}$ is the total number of instances in the dataset, from which $N_{data\_positive}$ and $N_{data\_negative}$ are the number of positive instances and the number of negative instances within the dataset. ACC measures the performance of a model when the data are balanced, while SENS and SPEC are more specific for unbalanced problems.
(3)$$ACC=\frac{TP+TN}{TP+TN+FP+FN}=\frac{TP+TN}{N_{data}}$$
(4)$$SENS=\frac{TP}{TP+FN}=\frac{TP}{N_{data\_positive}}$$
(5)$$SPEC=\frac{TN}{TN+FP}=\frac{TN}{N_{data\_negative}}$$

Furthermore, to compare more directly the effects of the different parameter values and find the most-optimal configuration, the Receiver Operating Characteristic (ROC) curves and their associated Area Under Curve (AUC) values were used. The ROC curve is a representation of the proportion between the True Positive Rate and the False Positive Rate of the classification, and each curve has its own AUC value in the range [0, 1]. When comparing models using the ROC curve, the closer the curve of the model passes through the perfect classification point [FPR = 0, TPR = 1], the higher the AUC value and the better the model are. Only ROC curves corresponding to the global performance of the system were computed.

## 3. Results and Discussion

Let us remember that the classification results were obtained after training each model with each training fold and evaluating with two different test sets: a test set created before applying DA (i.e., available real data; it is called Test 1) and the test formed after applying DA (i.e., simulating different cases that are not contemplated until now in the real data; it is called Test 2) (see Figure 9). We compared whether introducing Test 2 allowed us not only to improve the already tested classifiers, but also to obtain or to discriminate between models, so a better model selection can be performed.

The complete PPR detection performance results of both 2C-KNN and DL-NN techniques are shown in Table 2 and Table 3, respectively: each table shows the results using Test 1 (without DA) in the upper half, while the lower half shows the results using Test 2 (with DA).

The best version of each classifier was the one with the parameter *K* fixed at three neighbors in the case of 2C-KNN and the one with the parameter *n* fixed at 10 hidden neurons for the DL-NN. Their performances are included in Table 4: the boxplots (see Figure 10) and ROC curves (see Figure 11 display a comparison between the performance results before and after the application of DA, allowing the analysis of the effects of the DA procedure on the results for both classifiers. The application of the DA step to increase the PPR instances and balance the dataset reduced the variance of the performance values, which means that it managed to make both classifiers more robust.

As can be seen, the ACC and SPEC measurements reached very high values: around 95% for 2C-KNN, while those from the DL-NN were around 98%. In terms of SENS, 2C-KNN and the DL-NN reached values around 75% and 85%, respectively. By visually comparing these results, it is clear that the best PPR detection technique was the DL-NN method with 10 hidden neurons.

Results depicted in the boxplots and tables show the DL-NN models outperforming the 2C-KNN. However, the differences are smaller when comparing the performance of each type of model with the two experimentation setups Test 1 and Test 2. For this comparison, we decided to use the SENS measurement because it is the performance metric with worse results (smaller values and higher dispersion), testing whether results from both experimentation setups belong to the same population with the same probability distribution. We need a hypothesis contrast test to determine whether or not the designed experimentation with DA allows for obtaining better models. We ran the Shapiro normality test and the Levene homogeneity test to determine if the sub-populations followed a normal distribution and if they had the same variance; according to our results, all the sub-populations were normally distributed and presented the same variance. In this case, the most-suitable hypothesis test is the parametric version of the Analysis Of Variance (ANOVA) test [51]; we ran this test with a significance value of 95%.

We applied the following ANOVA test:To determine if all the 2C-KNN models obtained using Test 1 belong to the same population or not.To determine if all the 2C-KNN models trained using Test 2 belong to the same population or not.To determine if all the DL-NN models obtained using Test 1 belong to the same population or not.To determine if all the DL-NN models trained using Test 2 belong to the same population or not.

Results from the ANOVA tests showed that the 2C-KNN models obtained using the Test 1 experimentation belonged to the same distribution, making it impossible to decide which parameter set is the best. On the other hand, the results from 2C-KNN using the Test 2 experimentation performed differently: the ANOVA test rejected the null hypothesis that all of them belong to the same population. Therefore, for the 2C-KNN, the Test 2 experimentation setup allowed us to choose a better model and parameter set. On the other hand, the ANOVA test did not reject the null hypothesis for the DL-NN models: the Test 2 experimentation setup did not differentiate the performance of the parameter subsets.

In this latter case of the DL-NN models, the performance did not improve independently of the amount of available data or the network size. No over-fitting was detected, so we can conclude that we reached the performance bound for the evaluated DL-NN models given the available data. More complex DL structures and, consequently, more representative data are needed to improve these results.

Unfortunately, we cannot compare against other approaches in the literature. As mentioned before, the single study found so far on PPR detection was our previous research shown in [16], which focused on Type-4 PPR detection; Figure 12 shows the obtained results from our previous study. A comparison between the results from the two pieces of research shows that the SENS measurement was similar in both cases; however, the ACC and SPEC measurements were much worse in [16] than in this research. This fact reinforces our hypothesis that detecting PPRs as a single unique label produces a better performance of the models. In the same way, there is a significant increase in the performance of the models using the experimental setup described in this research.

## 4. Conclusions

For this study, our research team proposed incorporating a DA stage in a previously developed automatic PPR detection procedure [15,16] to solve our dataset imbalance problem. A Cross-Validation-based scheme with an additional DA step was applied in order to increase the number of PPR windows (the minority class, which occupies a total of 3% of the total number of instances in the dataset) and undersample the excess of non-PPR windows within each fold, thus balancing the training set.

Raw EEG signals were windowed and labeled before the DA technique was applied to artificially create new raw PPR windows that resemble the real ones as much as possible. Once the synthetic data were created, EEG windows were preprocessed and dimensionally reduced by extracting a total of 32 features from different domains and applying the PCA algorithm to reduce them into an even smaller set of 12 uncorrelated components. The final data were used for training and evaluating the ML models selected for the detection task: 2C-KNN and DL-NN.

This process was designed to detect all PPR types, contrary to our previous study, which was focused on the detection of only Type-4 PPRs [16]. DA allowed us to achieve better detection results compared to those previously obtained despite the incorporation of higher morphological variability due to their quite different waveforms, making the models more robust. Both methods performed very well, reaching average values of the ACC and SENS around 95% for 2C-KNN and above 98% for the DL-NN. However, SENS measurement showed that the DL-NN method was better than the 2C-KNN algorithm with average values of almost 85% compared to the average of 75% for 2C-KNN.

This detection performance was achieved by using simple ML techniques. For future work, DL models will start to be tested for this purpose due to their increased ability to learn the different complex PPR patterns more easily. Moreover, it is likely that, in the case of the DL-NN, integrating the DA step as an additional layer of the neural network and analyzing different data generator models can lead to better and more robust models. However, due to the high imbalance of the data, it may be convenient to approach the problem as a one-class problem oriented toward anomaly detection.

## Figures and Tables

**Figure 1 sensors-23-02312-f001:**
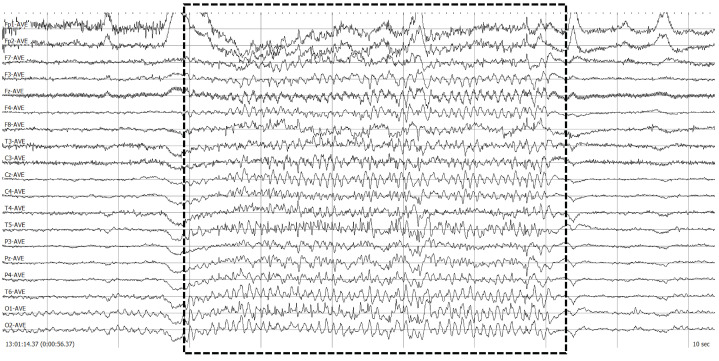
Example of EEG signals for PPRs when flashing a patient. The central area delimited with a dashed rectangle represents a PPR.

**Figure 2 sensors-23-02312-f002:**
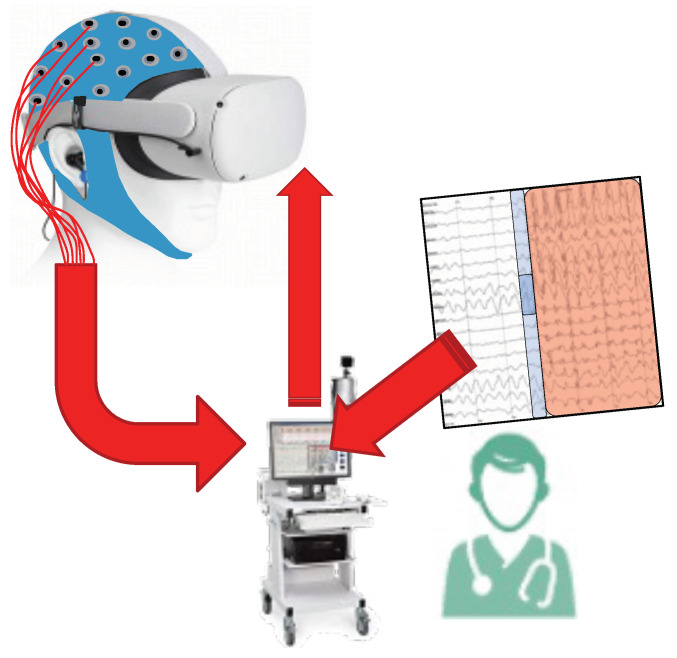
vAI4Neuron project overview. EEG or magnetoencephalograms can be used for monitoring the brain activity. VR is in charge of the stimulus, while AI assists the experts in the decision.

**Figure 3 sensors-23-02312-f003:**
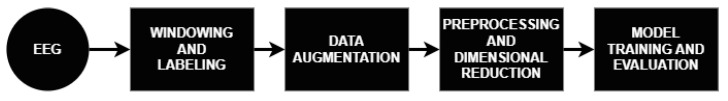
Workflow of this research. The original EEG data from the dataset are windowed and labeled. The Cross-Validation introduces a subsampling for reducing the number of non-PPR windows, while DA takes place to balance the number of PPR windows in the training/testing dataset. Window preprocessing, feature extraction, and dimensional reduction are applied to each EEG window. Finally, the process ends with the training and testing of the models.

**Figure 4 sensors-23-02312-f004:**
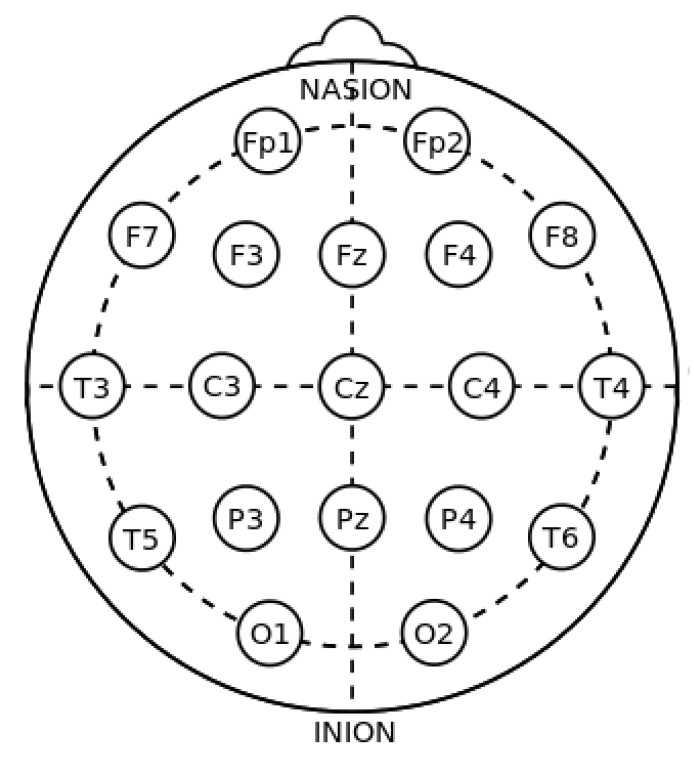
Position of the 19 electrodes following the 10–20 international standardized system, where “Nasion” is the point located at the center of the frontonasal area and “Inion” is the point located at the central point on the back of the neck.

**Figure 5 sensors-23-02312-f005:**
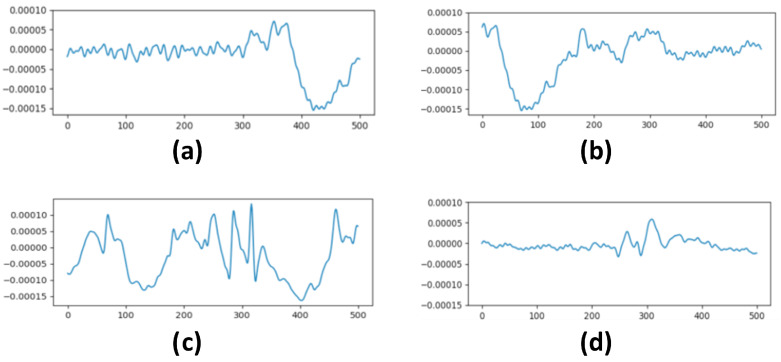
Different cases of a sliding window including a PPR event: (**a**) including only the starting part of a PPR (PPR onset); (**b**) including only the ending part of a PPR (PPR offset); (**c**) including the middle part of a PPR signal (fully covered window); (**d**) including the whole PPR (start and end).

**Figure 6 sensors-23-02312-f006:**
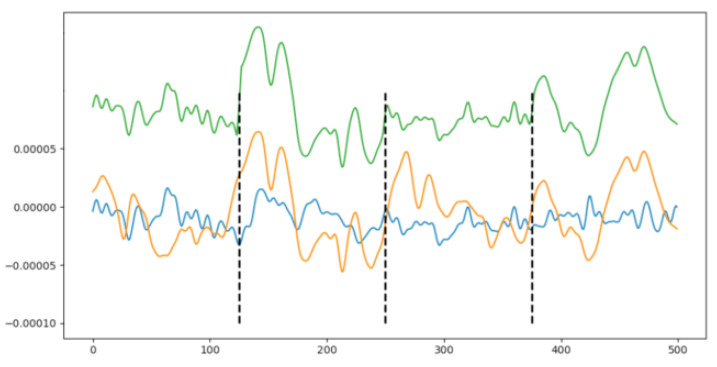
Example of the DA technique. A newly generated synthetic realistic PPR window (green signal) using two fully covered windows: **A** (blue) and **B** (orange). Dotted lines represent the cut-points where the signals were split.

**Figure 7 sensors-23-02312-f007:**
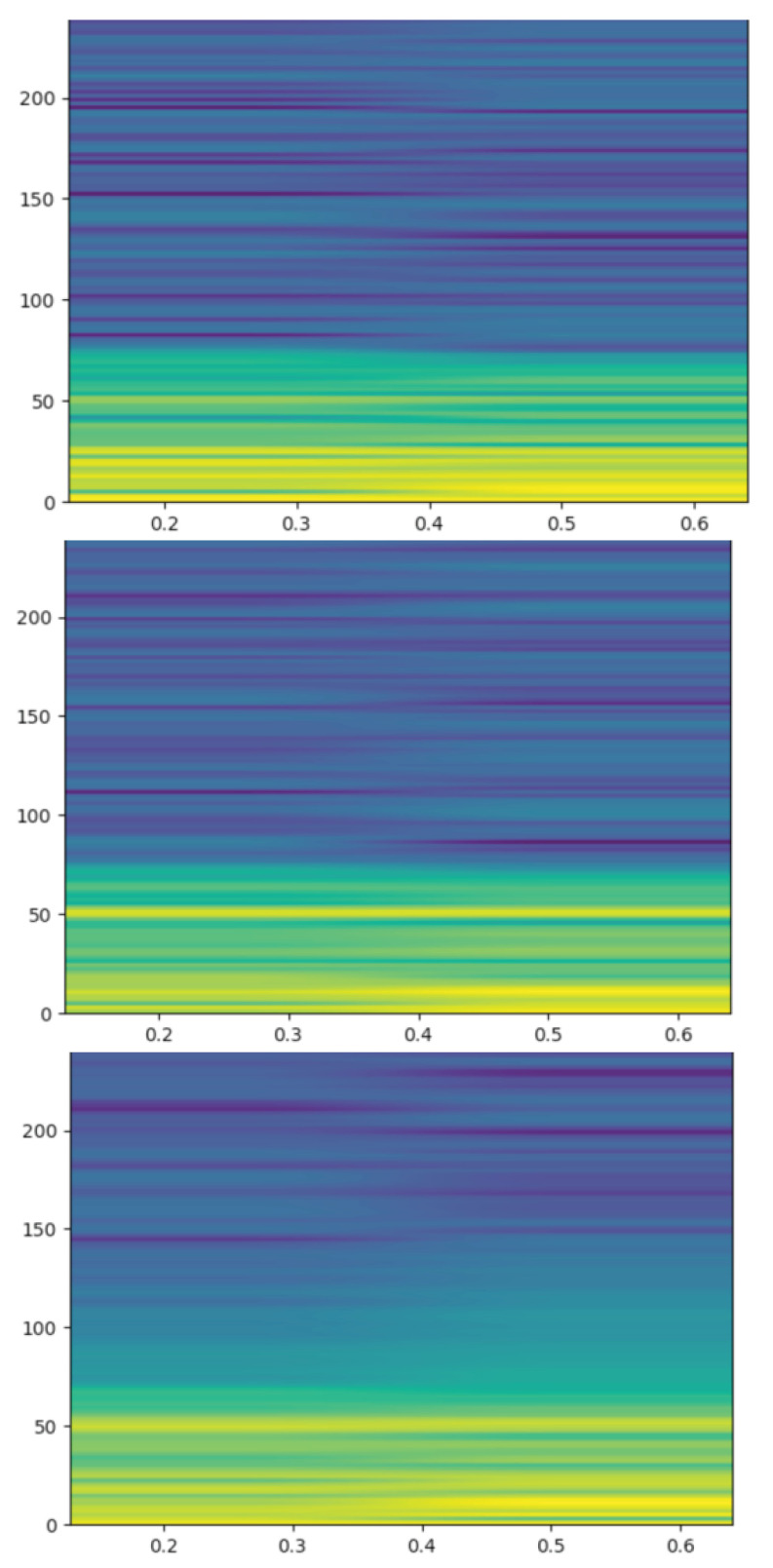
Spectrograms from a DA generation. Images at the top and in the center show the spectrograms of the two PPR windows acting as parents, while the image at the bottom depicts the spectrogram from the synthetic PPR window. The x-axis represents time, while the y-axis shows frequency.

**Figure 8 sensors-23-02312-f008:**
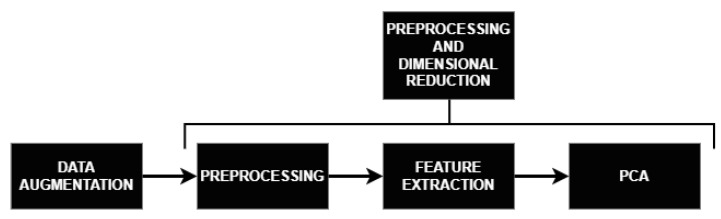
Workflow for the preprocessing and dimensional reduction stages. Firstly, the dataset was augmented, introducing new EEG windows. Afterwards, the preprocessing, feature extraction, and feature reduction stages were applied in a sequence on every EEG window (either being augmented or the original).

**Figure 9 sensors-23-02312-f009:**
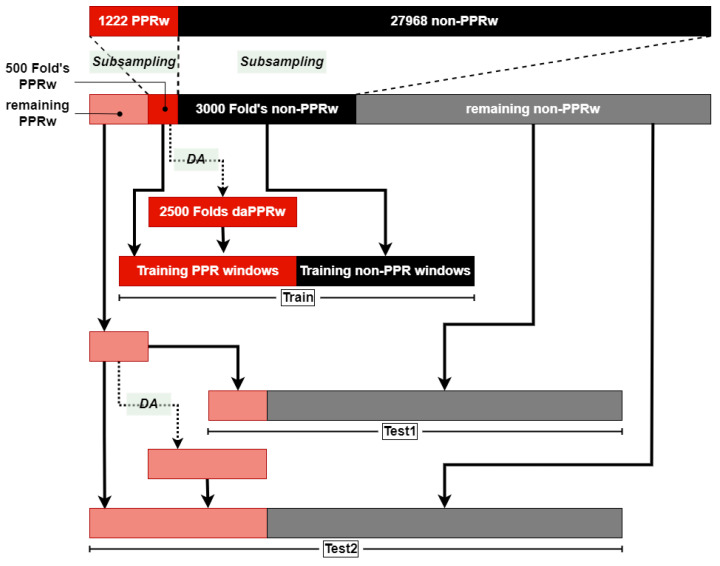
Cross-Validation data generation for each repetition. The training set includes 500 PPRw plus 2500 data-augmented PPRw (daPPRw) plus 3000 non-PPR windows (non-PPRw); PPRw and non-PPRw were subsampled from the original dataset. Two test datasets (Test 1 and Test 2) were created: with or without DA. Test 1 included all the remaining instances from the original dataset (722 PPRw and 24,968 non-PPRw), while Test 2 also included (3000-722) daPPRw.

**Figure 10 sensors-23-02312-f010:**
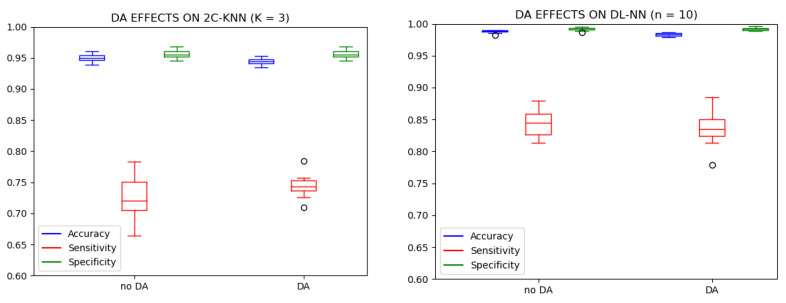
Boxplots from the best version of each classifier using both test sets. The **left** figure corresponds to the 2C-KNN version with parameter *K* fixed at 3 neighbors, and the **right** figure corresponds to the DL-NN version with parameter *n* fixed at 10 hidden neurons.

**Figure 11 sensors-23-02312-f011:**
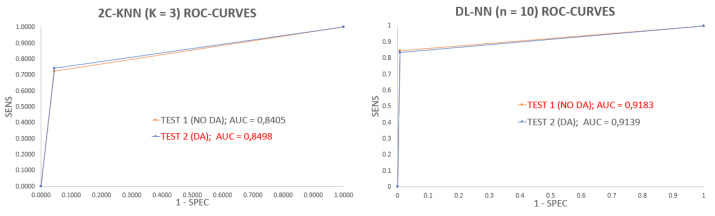
ROC curves from the best version of each classifier using both test sets. The **left** figure corresponds to the 2C-KNN version with parameter *K* fixed at 3 neighbors, and the **right** figure corresponds to the DL-NN version with parameter *n* fixed at 10 hidden neurons.

**Figure 12 sensors-23-02312-f012:**
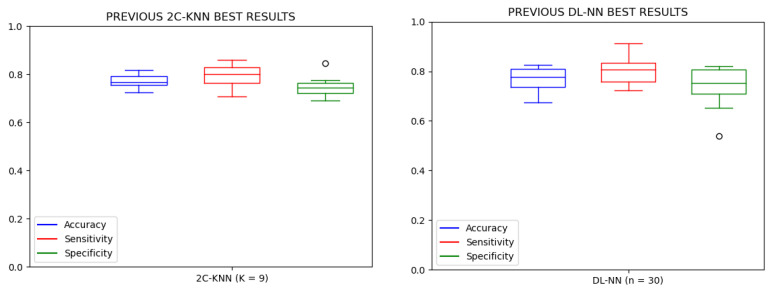
Results from the Type-4 PPR detection experiment performed in [16]. On the **left**, the boxplot of the results for the 2C-KNN technique with *K* fixed at 9 neighbors. On the **right**, the boxplot of the results for the DL-NN method with the parameter *n* fixed at 20 hidden neurons.

**Table 1 sensors-23-02312-t001:** Transformations of the EEG windows.

Statistical Domain
Kurtosis [45], Skewness [45], Standard Deviation, Maximum,
Minimum, Mean of Absolute Deviation, Root Mean Square.
**Temporal Domain**
Sum of Absolute Values, Maximum Amplitude, Sum of Absolute Differences,
Total Energy, Absolute Energy, Area Under the Curve, Entropy, Autocorrelation.
**Spectral Domain**
Fundamental Frequency, Maximum Frequency, Median Frequency, Maximum Power
Spectrum [46], Spectral Centroid [47], Spectral Decrease [47], Spread [47],
Spectral Distance, Spectral Kurtosis [47], Spectral Skewness [47], Spectral Entropy [48],
Positive Turning Points, Roll-Off [46], Roll-On [46], Variation [47],
Power Bandwidth [49], Human Range Energy [50].

**Table 2 sensors-23-02312-t002:** Results obtained for the 2C-KNN with the two experimentation setups. The upper part shows the figures obtained with the Test 1 experimentation (test data without including extra DA data), while the bottom part shows the results with the Test 2 experiment. $P_{i}$ refers to participant i, with i varying from 1 to 10.

	K = 3			K = 5			K = 7			K = 9		
$P_{i}$	ACC	SENS	SPEC	ACC	SENS	SPEC	ACC	SENS	SPEC	ACC	SENS	SPEC
$P_{1}$	0.9521	0.6639	0.9603	0.9647	**0.6681**	0.9731	0.9699	0.6639	0.9786	0.9728	0.6610	0.9817
$P_{2}$	0.9464	**0.7496**	0.9520	0.9569	0.7398	0.9631	0.9616	0.7398	0.9679	0.9660	0.7271	0.9728
$P_{3}$	0.9607	**0.7159**	0.9677	0.9686	0.7032	0.9761	0.9750	0.6990	0.9828	0.9783	0.6962	0.9863
$P_{4}$	0.9541	**0.7511**	0.9599	0.9657	0.7215	0.9727	0.9705	0.7342	0.9772	0.9729	0.7145	0.9803
$P_{5}$	0.9502	**0.7243**	0.9567	0.9599	0.7117	0.9669	0.9659	0.7103	0.9732	0.9695	0.7117	0.9769
$P_{6}$	0.9492	**0.7834**	0.9539	0.9579	0.7722	0.9632	0.9638	0.7623	0.9695	0.9690	0.7665	0.9748
$P_{7}$	0.9543	**0.7032**	0.9614	0.9641	0.7018	0.9716	0.9687	0.6906	0.9766	0.9726	0.6934	0.9805
$P_{8}$	0.9394	**0.7103**	0.9459	0.9523	0.7131	0.9591	0.9577	0.6948	0.9652	0.9620	0.6779	0.9701
$P_{9}$	0.9426	**0.7637**	0.9477	0.9556	0.7426	0.9617	0.9602	0.7370	0.9666	0.9651	0.7229	0.972
$P_{10}$	0.9463	**0.6864**	0.9537	0.9553	0.6624	0.9636	0.9612	0.6667	0.9696	0.9641	0.6667	0.9725
mean	0.9495	0.7252	0.9559	0.9601	0.7136	0.9671	0.9654	0.7098	0.9727	0.9692	0.7038	0.9768
median	0.9497	0.7201	0.9553	0.9589	0.7124	0.9653	0.9648	0.7046	0.9714	0.9693	0.7039	0.9758
std	0.0062	0.0368	0.0066	0.0054	0.0333	0.0058	0.0055	0.0328	0.0059	0.0050	0.0317	0.0052
	K = 11			K = 13			K = 15					
$P_{i}$	ACC	SENS	SPEC	ACC	SENS	SPEC	ACC	SENS	SPEC			
$P_{1}$	0.9739	0.6498	0.9832	0.9752	0.6456	0.9846	**0.9764**	0.6414	**0.9859**			
$P_{2}$	0.9683	0.7201	0.9753	0.9696	0.7103	0.9770	**0.9707**	0.7060	**0.9783**			
$P_{3}$	0.9794	0.6821	0.9878	0.9802	0.6864	0.9886	**0.9810**	0.6779	**0.9896**			
$P_{4}$	0.9763	0.7173	0.9837	0.9777	0.7131	0.9852	**0.9787**	0.7089	**0.9864**			
$P_{5}$	0.9725	0.7089	0.9800	0.9742	0.7060	0.9819	**0.9750**	0.6948	**0.9829**			
$P_{6}$	0.9730	0.7595	0.9791	0.9752	0.7581	0.9814	**0.9773**	0.7511	**0.9837**			
$P_{7}$	0.9741	0.6821	0.9825	0.9756	0.6751	0.9842	**0.9771**	0.6821	**0.9855**			
$P_{8}$	0.9642	0.6765	0.9724	0.9664	0.6779	0.9746	**0.9674**	0.6765	**0.9757**			
$P_{9}$	0.9671	0.7201	0.9741	0.9681	0.7131	0.9754	**0.9707**	0.7117	**0.9781**			
$P_{10}$	0.9663	0.6624	0.9749	0.9683	0.6540	0.9773	**0.9698**	0.6498	**0.9789**			
mean	0.9715	0.6979	0.9793	0.9731	0.6940	0.9810	0.9744	0.6900	0.9825			
median	0.9728	0.6955	0.9796	0.9747	0.6962	0.9816	0.9757	0.6885	0.9833			
std	0.0048	0.0330	0.0050	0.0046	0.0331	0.0047	0.0045	0.0320	0.0045			
	K = 3			K = 5			K = 7			K = 9		
$P_{i}$	ACC	SENS	SPEC	ACC	SENS	SPEC	ACC	SENS	SPEC	ACC	SENS	SPEC
$P_{1}$	0.9470	**0.7260**	0.9603	0.9587	0.7190	0.9731	0.9635	0.7120	0.9786	0.9659	0.7033	0.9817
$P_{2}$	0.9410	**0.7573**	0.9520	0.9507	0.7433	0.9631	0.9550	0.7410	0.9679	0.9593	0.7333	0.9728
$P_{3}$	0.9531	**0.7100**	0.9677	0.9604	0.6983	0.9761	0.9662	0.6897	0.9828	0.9689	0.6790	0.9863
$P_{4}$	0.9476	**0.7437**	0.9599	0.9588	0.7270	0.9727	0.9627	0.7207	0.9772	0.9649	0.7097	0.9803
$P_{5}$	0.9453	**0.7553**	0.9567	0.9542	0.7433	0.9669	0.9600	0.7400	0.9732	0.9632	0.7360	0.9769
$P_{6}$	0.9443	**0.7840**	0.9539	0.9527	0.7770	0.9632	0.9578	0.7627	0.9695	0.9627	0.7620	0.9748
$P_{7}$	0.9490	**0.7417**	0.9614	0.9579	0.7303	0.9716	0.9620	0.7183	0.9766	0.9656	0.7170	0.9805
$P_{8}$	0.9345	**0.7450**	0.9459	0.9464	0.7353	0.9591	0.9515	0.7227	0.9652	0.9557	0.7170	0.9701
$P_{9}$	0.9357	**0.7350**	0.9477	0.9486	0.7300	0.9617	0.9528	0.7230	0.9666	0.9576	0.7180	0.9720
$P_{10}$	0.9415	**0.7393**	0.9537	0.9506	0.7343	0.9636	0.9554	0.7197	0.9696	0.9581	0.7180	0.9725
mean	0.9439	0.7437	0.9559	0.9539	0.7338	0.9671	0.9587	0.7250	0.9727	0.9622	0.7193	0.9768
median	0.9448	0.7427	0.9553	0.9535	0.7323	0.9653	0.9589	0.7217	0.9714	0.9630	0.7175	0.9758
std	0.0058	0.0197	0.0066	0.0049	0.0200	0.0058	0.0049	0.0195	0.0059	0.0043	0.0218	0.0052
	K = 11			K = 13			K = 15					
$P_{i}$	ACC	SENS	SPEC	ACC	SENS	SPEC	ACC	SENS	SPEC			
$P_{1}$	0.9672	0.7010	0.9832	0.9682	0.6947	0.9846	**0.9693**	0.6923	**0.9859**			
$P_{2}$	0.9615	0.7310	0.9753	0.9626	0.7227	0.9770	**0.9637**	0.7217	**0.9783**			
$P_{3}$	0.9698	0.6703	0.9878	0.9706	0.6707	0.9886	**0.9713**	0.6677	**0.9896**			
$P_{4}$	0.9678	0.7037	0.9837	0.9691	0.7013	0.9852	**0.9703**	0.7023	**0.9864**			
$P_{5}$	0.9657	0.7280	0.9800	0.9673	0.7250	0.9819	**0.9680**	0.7193	**0.9829**			
$P_{6}$	0.9667	0.7607	0.9791	0.9688	0.7580	0.9814	**0.9708**	0.7550	**0.9837**			
$P_{7}$	0.9671	0.7117	0.9825	0.9688	0.7120	0.9842	**0.9699**	0.7100	**0.9855**			
$P_{8}$	0.9580	0.7177	0.9724	0.9600	0.7160	0.9746	**0.9608**	0.7137	**0.9757**			
$P_{9}$	0.9596	0.7183	0.9741	0.9605	0.7130	0.9754	**0.9630**	0.7113	**0.9781**			
$P_{10}$	0.9602	0.7147	0.9749	0.9619	0.7060	0.9773	**0.9632**	0.7017	**0.9789**			
mean	0.9644	0.7157	0.9793	0.9658	0.7119	0.9810	0.9670	0.7095	0.9825			
median	0.9662	0.7162	0.9796	0.9677	0.7125	0.9816	0.9687	0.7107	0.9833			
std	0.0041	0.0232	0.0050	0.0040	0.0225	0.0047	0.0039	0.0223	0.0045			

**Table 3 sensors-23-02312-t003:** Results obtained for the DL-NN with the two experimentation setups. The upper part shows the figures obtained with the Test 1 experimentation (test data without including extra DA data), while the bottom part shows the results with the Test 2 experiment. $P_{i}$ refers to participant i, with i varying from 1 to 10.

	n = 10			n = 20			n = 30		
$P_{i}$	ACC	SENS	SPEC	ACC	SENS	SPEC	ACC	SENS	SPEC
$P_{1}$	0.9871	**0.8467**	0.9911	0.9859	0.8326	0.9903	0.9893	0.8312	0.9938
$P_{2}$	**0.9903**	**0.8284**	**0.9950**	0.9861	0.8172	0.9909	0.9836	0.7947	0.9890
$P_{3}$	0.9879	**0.8594**	0.9916	0.9844	**0.8594**	0.9879	0.9892	0.8312	0.9937
$P_{4}$	0.9820	0.8256	0.9865	0.9885	0.8242	0.9932	0.9879	0.8340	0.9923
$P_{5}$	**0.9893**	0.8129	**0.9944**	0.9879	0.8256	0.9926	0.9887	0.8200	0.9936
$P_{6}$	0.9851	0.8439	0.9891	0.9827	0.8425	0.9867	0.9851	**0.8565**	0.9887
$P_{7}$	0.9899	**0.8790**	0.9930	0.9883	0.8523	0.9922	0.9889	0.8551	0.9928
$P_{8}$	0.9877	0.8158	0.9926	0.9864	0.8101	0.9914	**0.9890**	0.8129	**0.9940**
$P_{9}$	0.9892	0.8579	0.9929	0.9877	0.8523	0.9915	0.9908	**0.8608**	0.9946
$P_{10}$	0.9902	0.8776	0.9934	**0.9906**	0.8734	0.9939	0.9901	**0.8945**	0.9928
mean	0.9879	0.8447	0.9920	0.9868	0.8390	0.9911	0.9883	0.8391	0.9925
median	0.9886	0.8453	0.9928	0.9870	0.8376	0.9914	0.9890	0.8326	0.9932
std	0.0026	0.0239	0.0025	0.0023	0.0203	0.0023	0.0022	0.0284	0.0020
	n = 40			n = 50					
$P_{i}$	ACC	SENS	SPEC	ACC	SENS	SPEC			
$P_{1}$	0.9879	0.8312	0.9924	**0.9898**	0.8200	**0.9946**			
$P_{2}$	0.9896	0.8031	0.9949	0.9888	0.8214	0.9936			
$P_{3}$	**0.9919**	**0.8594**	**0.9957**	0.9812	0.8031	0.9862			
$P_{4}$	0.9893	**0.8481**	0.9934	**0.9905**	0.8298	**0.9950**			
$P_{5}$	0.9876	**0.8270**	0.9922	0.9872	0.8101	0.9922			
$P_{6}$	0.9851	0.8298	0.9895	**0.9891**	0.8158	**0.9941**			
$P_{7}$	0.9874	0.8565	0.9911	**0.9900**	0.8495	**0.9940**			
$P_{8}$	0.9875	**0.8214**	0.9923	0.9870	0.8186	0.9917			
$P_{9}$	**0.9919**	0.8383	**0.9962**	0.9879	0.8523	0.9917			
$P_{10}$	**0.9906**	0.8636	**0.9942**	0.9839	0.8819	0.9868			
mean	0.9889	0.8378	0.9932	0.9875	0.8302	0.9920			
median	0.9886	0.8347	0.9929	0.9883	0.8207	0.9929			
std	0.0022	0.0191	0.0021	0.0029	0.0240	0.0031			
	n = 10			n = 20			n = 30		
$P_{i}$	ACC	SENS	SPEC	ACC	SENS	SPEC	ACC	SENS	SPEC
$P_{1}$	0.9812	0.8453	0.9894	0.9835	**0.8463**	0.9917	0.9839	0.8447	0.9923
$P_{2}$	**0.9836**	0.7793	**0.9959**	0.9808	**0.7913**	0.9921	0.9797	0.7790	0.9918
$P_{3}$	0.9814	0.8277	0.9907	0.9791	0.8163	0.9888	0.9816	0.8050	0.9922
$P_{4}$	0.9787	0.8137	0.9886	0.9837	0.8263	0.9932	0.9802	0.8077	0.9906
$P_{5}$	0.9846	0.8397	**0.9933**	0.9840	**0.8480**	0.9922	0.9845	0.8383	**0.9933**
$P_{6}$	0.9808	**0.8310**	0.9897	0.9751	0.8140	0.9848	0.9783	0.8273	0.9873
$P_{7}$	**0.9852**	**0.8517**	0.9933	0.9831	0.8473	0.9912	0.9830	0.8423	0.9915
$P_{8}$	0.9827	0.8227	0.9923	0.9815	0.8200	0.9912	**0.9851**	0.8220	**0.9949**
$P_{9}$	0.9854	0.8630	0.9927	0.9854	0.8597	0.9930	**0.9879**	**0.8647**	0.9953
$P_{10}$	0.9866	**0.8847**	0.9928	0.9880	0.8787	**0.9945**	0.9844	0.8793	0.9907
mean	0.9830	0.8359	0.9919	0.9824	0.8348	0.9913	0.9829	0.8310	0.9920
median	0.9831	0.8353	0.9925	0.9833	0.8363	0.9919	0.9835	0.8328	0.9920
std	0.0025	0.0287	0.0022	0.0036	0.0257	0.0027	0.0029	0.0295	0.0023
$P_{i}$	ACC	SENS	SPEC	ACC	SENS	SPEC			
$P_{1}$	0.9827	0.8330	0.9917	**0.9847**	0.8437	**0.9932**			
$P_{2}$	0.9819	0.7700	0.9947	0.9824	0.7867	0.9942			
$P_{3}$	**0.9855**	**0.8290**	**0.9950**	0.9782	0.8113	0.9883			
$P_{4}$	**0.9842**	**0.8353**	0.9932	0.9826	0.7947	**0.9939**			
$P_{5}$	0.9830	0.8327	0.9920	**0.9847**	0.8437	0.9932			
$P_{6}$	0.9786	0.8043	0.9891	**0.9836**	0.8170	**0.9936**			
$P_{7}$	0.9810	0.8337	0.9898	0.9839	0.8247	**0.9935**			
$P_{8}$	0.9830	**0.8280**	0.9924	0.9815	0.8220	0.9911			
$P_{9}$	0.9877	0.8603	**0.9954**	0.9857	0.8613	0.9932			
$P_{10}$	**0.9876**	0.8717	**0.9945**	0.9839	0.8817	0.9900			
mean	0.9835	0.8298	0.9928	0.9831	0.8287	0.9924			
median	0.9830	0.8328	0.9928	0.9837	0.8233	0.9932			
std	0.0028	0.0278	0.0022	0.0021	0.0293	0.0020			

**Table 4 sensors-23-02312-t004:** Best performance results from both 2C-KNN with 3 neighbors and the DL-NN with 30 hidden neuron classifiers.

	2C-KNN → K = 3						DL-NN → n = 10					
	**Test 1 (No DA)**			**Test 2 (DA)**			**Test 1 (No DA)**			**Test 2 (DA)**		
** *P_i_* **	**ACC**	**SENS**	**SPEC**	**ACC**	**SENS**	**SPEC**	**ACC**	**SENS**	**SPEC**	**ACC**	**SENS**	**SPEC**
$P_{1}$	**0.9521**	0.6639	**0.9603**	0.9470	**0.7260**	**0.9603**	**0.9871**	**0.8467**	**0.9911**	0.9812	0.8453	0.9894
$P_{2}$	**0.9464**	0.7496	**0.9520**	0.9410	**0.7573**	**0.9520**	**0.9903**	**0.8284**	0.9950	0.9836	0.7793	**0.9959**
$P_{3}$	**0.9607**	**0.7159**	**0.9677**	0.9531	0.7100	**0.9677**	**0.9879**	**0.8594**	**0.9916**	0.9814	0.8277	0.9907
$P_{4}$	**0.9541**	**0.7511**	**0.9599**	0.9476	0.7437	**0.9599**	**0.9820**	**0.8256**	0.9865	0.9787	0.8137	**0.9886**
$P_{5}$	**0.9502**	0.7243	**0.9567**	0.9453	**0.7553**	**0.9567**	**0.9893**	0.8129	**0.9944**	0.9846	**0.8397**	0.9933
$P_{6}$	**0.9492**	0.7834	**0.9539**	0.9443	**0.7840**	**0.9539**	**0.9851**	**0.8439**	0.9891	0.9808	0.8310	**0.9897**
$P_{7}$	**0.9543**	0.7032	**0.9614**	0.9490	**0.7417**	**0.9614**	**0.9899**	**0.8790**	0.9930	0.9852	0.8517	**0.9933**
$P_{8}$	**0.9394**	0.7103	**0.9459**	0.9345	**0.7450**	**0.9459**	**0.9877**	0.8158	**0.9926**	0.9827	**0.8227**	0.9923
$P_{9}$	**0.9426**	**0.7637**	**0.9477**	0.9357	0.7350	**0.9477**	**0.9892**	0.8579	**0.9929**	0.9854	**0.8630**	0.9927
$P_{10}$	**0.9463**	0.6864	**0.9537**	0.9415	**0.7393**	**0.9537**	**0.9902**	0.8776	**0.9934**	0.9866	**0.8847**	0.9928

## Data Availability

Not applicable.

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
