# Peer review of "Data Augmentation Effects on Highly Imbalanced EEG Datasets for Automatic Detection of Photoparoxysmal Responses"

_sensors, 2023, doi:10.3390/s23042312_

Round 1
Reviewer 1 Report
My summary:
The paper presents a Data Augmentation (DA) approach to the brain diseases detection based on the EEG signals.
The Authors use a real-world EEG dataset; expand its content using certain selected DA technique; carry on the detection experiment; and finally, evaluate, analyze, and discuss the obtained empirical results.
Comments:
Even though, the work does not actually introduce a novel detection method, neither in the field of EEG signals detection, nor in the general area of machine learning, it still presents an undoubtedly valuable scientific contribution by conducting an original research focused on utilizing the DA solution for the benefit of the Photoparoxysmal Responses (PPRs) detection.
The idea of augmenting an EEG collection dataset sounds indeed, on one hand, interesting, and on the other hand, promising from the point of view of practical and experimental accuracy, effectiveness, and efficiency.
The problem of the PPR detection has been presented in the context of the current medical possibilities to struggle with it, which makes all the technical- and computer-science-oriented considerations even more justified, reasonable, valuable, and useful.
The experimental study has been carried out in a convincing and professional manner, ensuring the meaningful and reliable results.
However, I have the following doubts and concerns, which should be addressed, in my opinion, before the manuscript is ready to be published, in order to increase its clarity, comprehensibility, and easy reception by all the potential Readers.
Namely, I would like to mention about the following aspects, which are certain drawbacks, in my opinion:
1.) The structure of the paper might be improved.
I suggest to divide the Introduction of the paper, in a way that, most importantly, the original contribution of the Authors is clear and emphasized. The part, where it is stated that "This study is part of a larger project (...)" might be as well a subsection within the Introduction section. I think that this change would make more clear to the Reader, where the novelty and scientific value of the work lies, in fact.
Furthermore, the overview of the related work oughts to be moved to a separate section, and perhaps a bit extended.
2.) I would like also to suggest the Authors adding certain general description and explanation on the general issue of DA. The Authors use the sliding windows to artificially generate additional data for enlarging their original dataset. However, the Reader may be interested in general information about the framework of DA techniques. Certainly, a brief classification and taxonomy of the DA approaches could have been presented in this matter.
3.) Is there any particular reason for choosing the Principal Component Analysis (PCA) for the dimensionality reduction from 31 to 12 features? To me, 31 dimensions do not seem a high number of dimensions.
Besides, what is a reason of choosing PCA, in particular, instead of newer methods, like the category of Neighborhood-Preserving Projections (NPPs), to name but a few.
4.) Could the Authors provide a justification for the particular choice of the 2-class k-Nearest Neighbors (2ckNN) and the Neural Network with a Dense Layer as the hidden layer (DL-NN) as the classification methods in their PPR detection experimental research.
5.) My last remark concerns the notation within the presentation of the experimental results. In Tables 3 and 4, one can find notation F1, ..., F10. We do not see them all in Fig. 3. Frankly speaking, I feel a little confused, when reading Section 3, because of the notation utilized.
Assuming that all of the mentioned suggestions and concerns are properly addressed and the appropriate changes and corrections are introduced in the paper, I can consider the work as suitable for publication in the journal.
Author Response
Dear Reviewer,
Please, see the attached file, which contains our rebuttal letter.
Thank you for your review. We do expect the manuscript is now valid for publication.
Best wishes
The authors.

Reviewer 2 Report
Martins et al. aimed to develop a data augmentation method by generating synthetic EEG data of photoparoxysmal responses (PPR). The authors hypothesized that increasing the number of PPR EEG data and balancing the number of PPR and non-PPR EEG datasets would improve the machine-learning automated PPR detection performance. They studied scalp EEG data with PPR during photic stimulation in 10 patients with photosensitivity. They mixed the two 1-second lengths of PPR data to create synthetic PPR EEG data. They then generated training data with 500 raw PPR data, 2500 synthetic RRP data, and 3000 non-PPR data. They generated two test data: one included only raw PPR data, and another had both raw and synthetic PPR data. They found that the DL-NN method with 10 hidden neurons achieved a sensitivity of 85% and specificity and accuracy of 98% in both datasets. These performances were superior to those of the DL-NN method using raw type-4 PPR data in their previous study. The authors conclude that their data augmentation method could improve the performance of machine learning models in PPR detection.
Comments:
[1] Strength: The method is straightforward to reproduce.
[2] Weakness: Lacks direct comparison to traditional methods.
[3] Methods: It would be helpful to show a spectrogram of the synthetic EEG data to indicate whether the joints in the data have been properly smoothed. (Are the joints in the data showing artificial low-power or high-power bands?)
[4] Methods: If possible, please prove that the joints in the data described in [3] do not play a role in the machine learning performance (this is not required, as similar performance has been shown with data sets that do not include synthetic EEG data, although it would be more helpful to show this).
[5] Methods/Results: It would be helpful to create the training dataset using only raw EEG data and directly compare machine learning performance with the data augmentation method.
Author Response
Dear Reviewer,
Please, see the attached file containing our rebuttal letter.
Thank you for your review. We do expect the manuscript is now valid for publication.
Best wishes
The authors.

Reviewer 3 Report
This study used Data Augmentation technique in Photoparoxysmal Responses classification. The results show that DA is able to improve the models, making them more robust and more able to generalize. The idea is interesting. However, I think the paper needs substantial modification before it can be published.
1, Introduction: Usually, there is no need to put figures in the introduction part., And the figure quality is super bad.
2, The dataset: no subject information can be found. What is the age, and gender?
3, The reason for choosing the single lead Fz is not clear.
4, In Line 219, why are the features normalized between 0~1? It is not commonly seen.
5, In line 225 to 236, How the number of non-PPR instances is reduced is not clear.
6, The result part is hard to understand
Author Response

(The authors gave the same response as above.)

Round 2
Reviewer 3 Report
The authors solved the issues. I have no more questions.
Author Response
Dear Reviewer,
Thank you for these comments.
According to this review, no changes were needed. This new document release has been enriched with suggestions from the Academic Editor.
Best wishes!
The authors:
Fernando Moncada
José R. Villar
Beatriz García-López
Víctor M. González